# *Lactobacillus bulgaricus* or *Lactobacillus rhamnosus* Suppresses NF-κB Signaling Pathway and Protects against AFB_1_-Induced Hepatitis: A Novel Potential Preventive Strategy for Aflatoxicosis?

**DOI:** 10.3390/toxins11010017

**Published:** 2019-01-04

**Authors:** Yuanyuan Chen, Ruirui Li, Qiaocheng Chang, Zhihao Dong, Huanmin Yang, Chuang Xu

**Affiliations:** 1College of Animal Science and Veterinary Medicine, Heilongjiang Bayi Agricultural University, Daqing High-Tech Industrial Development Zone, Daqing 163319, China; chenyuanyuan2001@hotmail.com (Y.C.); Liruirui0507@hotmail.com (R.L.); changqiaocheng2018@hotmail.com (Q.C.); dzh18346662886@hotmail.com (Z.D.); 2State Key Laboratory of Pathogen and Biosecurity, Beijing Institute of Microbiology and Epidemiology, Fengtai District, Beijing 100071, China

**Keywords:** aflatoxin B_1_, liver, *Lactobacillus bulgaricus*, *Lactobacillus rhamnosus*, NF-κB, inflammation

## Abstract

Aflatoxin B_1_ (AFB_1_), a mycotoxin found in food and feed, is immunotoxic to animals and poses significant threat to the food industry and animal production. The primary target of AFB_1_ is the liver. To overcome aflatoxin toxicity, probiotic-mediated detoxification has been proposed. In the present study, to investigate the protective effects and molecular mechanisms of *Lactobacillus bulgaricus* or *Lactobacillus rhamnosus* against liver inflammatory responses to AFB_1_, mice were administered with AFB_1_ (300 μg/kg) and/or *Lactobacillus* intragastrically for 8 weeks. AML12 cells were cultured and treated with AFB_1_, BAY 11-7082 (an NF-κB inhibitor), and different concentrations of *L. bulgaricus* or *L. rhamnosus.* The body weight, liver index, histopathological changes, biochemical indices, cytokines, cytotoxicity, and activation of the NF-κB signaling pathway were measured. AFB_1_ exposure caused changes in liver histopathology and biochemical functions, altered inflammatory response, and activated the NF-κB pathway. Supplementation of *L. bulgaricus* or *L. rhamnosus* significantly prevented AFB_1_-induced liver injury and alleviated histopathological changes and inflammatory response by decreasing NF-κB p65 expression. The results of in vitro experiments revealed that *L.*
*rhamnosus* evidently protected against AFB_1_-induced inflammatory response and decreased NF-κB p65 expression when compared with *L. bulgaricus*. These findings indicated that AFB_1_ exposure can cause inflammatory response by inducing hepatic injury, and supplementation of *L. bulgaricus* or *L. rhamnosus* can produce significant protective effect against AFB_1_-induced liver damage and inflammatory response by regulating the activation of the NF-κB signaling pathway.

## 1. Introduction

Aflatoxins (AFs) are secondary metabolites produced by *Aspergillus* spp., mainly *Aspergillus flavus* and *Aspergillus parasiticus* [1]. Among the AFs, aflatoxin B_1_ (AFB_1_) is the most toxic AF known so far, and has been regarded as a class I carcinogen by WHO since 1993 [2]. It has been reported that the toxicity of AFB_1_ corresponds to 10 times that of potassium cyanide and 68 times that of arsenic [3]. Currently, more than 20 kinds of AFs have been identified, including AFB_1_, B_2_, G_1_, and G_2_, of which AFB_1_ is the most widely studied [4]. In China, AF contamination generally refers to AFB_1_ contamination. AFB_1_ causes significant damage and loss, and is gaining increasing attention because of its toxicity, carcinogenicity, and universality [5,6].

In recent years, mold contamination of feedstuffs and compound feed is becoming increasingly serious in China. The high detection rate of mycotoxin brings significant threat to animal husbandry, and to a certain extent, restricts the development of trade of agricultural and animal husbandry products. As AF accumulation is hazardous to human health and environment security through food chains, exploration of effective methods and mechanisms of AFB_1_ prevention has important economic and social significance.

AFB_1_ affects the internal organs of humans and animals, especially the liver, and ingestion of a certain amount of AFB_1_ can result in acute poisoning, including acute hepatitis, hemorrhagic necrosis, steatosis, and bile canaliculi proliferation [7]. The liver is the main site of AFB_1_ metabolism, and AFB_1_ poisoning has been noted to cause overexpression of inflammatory cytokines of liver in rats. A variety of inflammatory cytokines that influence hepatocytes have been identified, among which tumor necrosis factor-alpha (TNF-α), interleukin-1 (IL-1), interleukin-6 (IL-6), and interleukin-8 (IL-8) are important [8]. As excessive or uncontrolled expression of these cytokines can promote inflammatory reaction and liver injury [9,10], the degree of hepatic damage could be reduced by inhibiting the overexpression of inflammatory cytokines. Thus, inhibition of inflammatory factors may be a novel way to improve the prevention and treatment of AFB_1_ poisoning. Our previous studies have shown that lactic acid bacteria could alleviate the degree of liver inflammation and hepatic damage, and exert a positive effect on the prevention of AFB_1_ poisoning in chickens, which are one of the most susceptible to AFB_1_ poisoning. Nevertheless, further studies are needed to determine whether lactic acid bacteria could produce the same effect on the prevention of AFB_1_ poisoning in mammals, as well as investigate the mechanism underlying the regulation of inflammatory factors by lactic acid bacteria to alleviate AFB_1_ poisoning. Accordingly, control of inflammation has become a new strategy for the treatment of conditions caused by inflammatory cells based on the NF-κB signal pathway.

NF-κB, which occurs in a variety of cells, is an important nuclear transcription factor in signal pathways and is associated with inflammation. When the tissue cells are invaded by bacteria, viruses, or fungi, the immune responses are initiated, thereby leading to the activation of NF-κB [11,12,13]. Activated NF-κB plays a vital role in host defense and inflammatory response by regulating multiple cytokines and is indirectly involved in inflammatory reactions [14]. In recent years, various cell signaling pathways, including NF-κB, ERK, PKB, and MAPKs, have been examined, among which the NF-κB signaling pathway is the major regulatory pathway [15,16,17]. Lactic acid bacteria and their metabolites regulate inflammatory cytokines by inhibiting some signaling pathways in the intestinal epithelial cells, thus exerting an anti-inflammatory effect [18]. A recent study indicated that probiotics exert protective effects on the liver [19], and oral administration of *Lactobacillus rhamnosus* GG (LGG) has been found to improve liver inflammation and intestinal dysfunction in a mouse model of alcoholic liver injury [20,21]. Besides, supplementation of VSL#3 in patients with alcoholic liver cirrhosis has been reported to decrease the expression of proinflammatory cytokines, significantly alleviate hepatopathy symptoms, and positively regulate intestinal function [22]. In addition, the NF-κB signaling pathway has been noted to be involved in hepatocytes inflammatory reaction [23,24]. Nevertheless, further research is required to determine whether lactic acid bacteria could also inhibit the secretion of inflammatory factors by regulating the NF-κB signaling pathway, and subsequently prevent inflammatory injury caused by AFB_1_. In particular, the following scientific problems need to be addressed:(1)Do lactic acid bacteria inhibit TNF-α, IL-1, IL-6, and IL-8 during inflammatory injury of the liver caused by AFB_1_?(2)Do lactic acid bacteria activate the NF-κB signaling pathway during inflammatory injury of the liver caused by AFB_1_?

In the present study, mouse and AFB_1_ cell injury models were established, and fluorescence qPCR and ELISA were used to detect the transcriptional and expression levels of inflammatory cytokines after the addition of lactic acid bacteria for alleviation of AFB_1_ poisoning. The expression of the NF-κB signaling pathway in hepatocytes and liver tissues was detected by Western blot analysis. As the regulatory mechanism of the NF-κB signaling pathway was examined both in vivo and in vitro, the findings of the present study could lay the foundation for research on the anti-inflammatory mechanism of lactic acid bacteria and provide a theoretical basis for the treatment of AFB_1_ poisoning using lactic acid bacteria in practice, as well as help in the development of lactic acid bacteria as feed additive for preventing AFB_1_ poisoning.

## 2. Results

### 2.1. Limited Effects of L. bulgaricus or L. rhamnosus on Body Weights and Liver Index

No mortality was recorded among all the experimental animals. As shown in Table 1, the body weight of mice in Group A was significantly higher (*p* < 0.05) on day 14 and lower (*p* > 0.05) from day 28 to day 56, when compared with that in Group C. The body weight of mice in Groups A + L1 and A + L2 was significantly lower (*p* < 0.05) on days 14 and 21 and on day 14, respectively, when compared with that in Group A. Furthermore, although there was no difference in the body weight of mice among the five groups on days 7, 28, 42, 49, and 56 (*p* > 0.05), the beneficial effects of *L. bulgaricus* on growth was better than those of *L. rhamnosus.* In addition, the liver index of mice treated with AFB_1_ was higher than that of mice in Group C (*p* < 0.05), whereas the liver index of mice treated with *L. bulgaricus* or *L. rhamnosus* was lower than that of the AFB_1_ group, but with no significance (*p* > 0.05).

### 2.2. Histopathological Observations of the Liver

In Group C, the livers showed regular hepatic sinusoids and normal hepatic cords (Figure 1a), and in Group D, no abnormalities or histological changes were observed in the livers (Figure 1b). In contrast, AFB_1_ exposure caused evident injury to the liver, including severe hepatic swelling and disappearing and disarranged structure of hepatic lobules (Figure 1c). Although AFB_1_ + *Lactobacillus* treatment groups exhibited similar significantly alleviated pathological changes, congestion spaces and inflammatory cell infiltration were still noted, especially in Group A + L2 (Figure 1d,e). These results showed that the effects of AFB_1_ on hepatic histological damage could be mitigated by addition of *L. bulgaricus* or *L. rhamnosus*.

### 2.3. L. bulgaricus or L. rhamnosus Alleviated AFB_1_-Induced Hepatic and Renal Dysfunction in Mice

The alterations in the biochemical indices of the five groups are shown in Figure 2. AST and ALT are perceived as useful markers of liver injury, and in the present study, the serum ALT and AST levels were significantly increased (*p* < 0.05) in the AFB_1_-treated group when compared with those in the control group. The levels of ALT and AST in Group A + L2 were significantly decreased (*p* < 0.05), whereas those in Group A + L1 were lower but did not show a significant difference (*p* > 0.05) when compared with the ALT and AST levels in Group A. In contrast, the ALP level did not exhibit marked changes among the five treatment groups. The levels of TP, BUN, and CREA were significantly decreased in Group A when compared with those in Group C (*p* < 0.05). In contrast, the serum TP, BUN, and CREA levels were slightly higher in Groups A + L1 and A + L2, when compared with those in Group A, whereas there were no significant differences in these indices between the AFB_1_ and/or *Lactobacillus* groups (*p* > 0.05).

### 2.4. L. bulgaricus or L. rhamnosus Reduced AFB_1_-Induced Inflammatory Response in Mice

Figure 3 illustrates the levels of various parameters related to inflammatory response in mice after different treatments. Mice treated with AFB_1_ showed significant upregulation of IL-1β (*p* < 0.05), TNF-α (*p* < 0.01), and IL-6 (*p* < 0.01) when compared with the control group. In contrast, *L. bulgaricus* and *L. rhamnosus* treatments downregulated serum IL-1β (*p* < 0.05), TNF-α (*p* < 0.05 and *p* < 0.01, respectively), and IL-6 (*p* < 0.01) levels in AFB_1_-induced mice. Furthermore, mice exposed to AFB_1_ exhibited decreased serum IFN-γ level when compared with mice in the control group (*p* < 0.05), whereas those subjected to *L. bulgaricus* or *L. rhamnosus* treatment presented an increase in IFN-γ level (*p* > 0.05). However, there was no difference in the IL-2 and IL-8 levels among all the treatment groups (*p* > 0.05).

### 2.5. L. bulgaricus or L. rhamnosus Effectively Downregulated the Expression of Hepatic Inflammatory Cytokines and NF-κB p65 Activation in AFB_1_-Induced Mice

To determine the effects of *L. bulgaricus* or *L. rhamnosus* on NF-κB expression in mice exposed to AFB_1_, the NF-κBp65, IL-1β, and TNF-α transcription and NF-κBp65 and IκBα protein expression were examined (Figure 4). AFB_1_ treatment induced an increase in the NF-κB p65 and TNF-α mRNA expression levels (*p* < 0.01) in the liver of mice. Conversely, *L. bulgaricus* or *L. rhamnosus* treatment regulated NF-κBp65 and IκBα gene and protein expression (*p* < 0.01), and the NF-κBp65 mRNA expression was significantly lower in Group A + L2 when compared with that in Group C (*p* < 0.01). A similar trend was also observed with respect to TNF-α mRNA expression. With regard to IL-1β mRNA expression, AFB_1_ exposure slightly increased IL-1β mRNA levels (*p* > 0.05); however, *L. bulgaricus* or *L. rhamnosus* treatment decreased the IL-1β mRNA levels, when compared with those in the AFB_1_-induced group (*p* < 0.01). Overall, these results indicated that *L. bulgaricus* or *L. rhamnosus* can decrease NF-κB p65 transcriptional activity in the liver.

### 2.6. Cytotoxicity of L. bulgaricus and L. rhamnosus to AML12 Cells

The toxic effects of 0–2.0 mg/mL *L. bulgaricus* and 0–5.0 mg/mL *L. rhamnosus* on the viability (Figure 5) of AML12 cells were concentration-dependent. Treatment with 1.4, 1.6, 1.8, and 2.0 mg/mL *L. bulgaricus* or 1.0, 2.0, 3.0, 4.0, and 5.0 mg/mL *L. rhamnosus* significantly decreased cell viability when compared with the control group (*p* < 0.05). Furthermore, 1.2 mg/mL *L. bulgaricus* and 0.5 mg/mL *L. rhamnosus* had no significant cytotoxic effects on AML12 cells. Therefore, 1.2 mg/mL *L. bulgaricus* and 0.5 mg/mL *L. rhamnosus* were used in the subsequent experiments.

### 2.7. L. bulgaricus and L. rhamnosus Exerted Varied Influences on the AFB_1_-Induced Inflammatory Response of AML12 Cells

It has been reported that overactivation of NF-κB pathways could promote the expression of inflammatory cytokines [25]. Accordingly, in the present study, the effects of overactivation of NF-κB pathways on inflammatory cytokines in cultured AML12 cells were examined (Figure 6). The concentrations of IL-1β and IL-8 were significantly higher in the AFB_1_-induced group when compared with those in the control group in 1, 2, 4, and 12 h (*p* < 0.05 and *p* < 0.01, respectively); however, administration of *L. rhamnosus* significantly decreased the concentrations of IL-1β and IL-8 when compared with those in the AFB_1_-induced group in different time periods. Similarly, the levels of TNF-α were significantly higher in the AFB_1_ treatment group when compared with those in the control group in 2, 4, and 12 h (*p* < 0.05 and *p* < 0.01, respectively), whereas administration of *L. rhamnosus* significantly decreased the levels of TNF-α when compared with those in the AFB_1_-treated group in 2 h. The concentrations of IL-6 were significantly lower in the AFB_1_-induced group when compared with those in the control group in 1, 2, 4, and 12 h (*p* < 0.01); however, *L. rhamnosus* administration significantly increased IL-6 concentrations when compared with those in the AFB_1_ treatment group in 1, 2, 4, and 12 h. The levels of IFN-γ were significantly higher in the AFB_1_ treatment group when compared with those in the control group in 1, 4, and 12 h (*p* < 0.05 and *p* < 0.01, respectively); however, no effective changes in the IFN-γ levels were observed in Groups A + L1 and A + L2 when compared with those in Group A. Besides, no significant changes in the levels of IL-2 were noted among the six groups in different time periods. These results indicated that AFB_1_ can induce the expression and synthesis of inflammatory cytokines in AML12 cells through overactivation of the NF-κB inflammatory pathway.

### 2.8. L. bulgaricus or L. rhamnosus Attenuated Upregulation of Inflammatory Cytokines and NF-κB p65 Activation in AFB_1_-Induced AML12 Cells

As shown in Figure 7, the IL-1β, TNF-α, and NF-κB p65 mRNA levels were increased in the AFB_1_ treatment group. However, *L. bulgaricus* effectively decreased AFB_1_-induced IL-1β, TNF-α, and NF-κB p65 levels in 4 h (*p* < 0.01), and *L. rhamnosus* significantly decreased AFB_1_-induced IL-1β levels in 4 h and TNF-α and NF-κB p65 levels in 4 and 12 h (*p* < 0.01). The NF-κB p65 protein expression was higher in Group A when compared with that in Group C in 4 h, but it decreased in Groups A + L1 or A + L2, when compared with that in Group A. In contrast, an opposite trend was noted with regard to IκBα protein expression, and no significant difference in NF-κB p65 and IκBα protein expression was detected among the six groups in 1 and 2 h. These results indicated that *L. bulgaricus* or *L. rhamnosus* can potentially regulate the NF-κB inflammatory pathway in AML12 cells.

## 3. Discussion

Mycotoxins widely occur in foods and are considered to pose food safety risks. In the present study, AFB_1_ was found to cause evident injury to the liver and significantly increase liver index and AST, ALT, IL-1β, and TNF-α levels, as well as reduce TP, BUN, CREA, and IFN-γ contents. These results indicated that AFB_1_ caused liver damage and inflammatory response in mice and AML12 cells. Recent studies have shown that certain probiotics have the ability to effectively protect from liver injury [26,27]. For example, *LGG* culture supernatant can ameliorate acute alcohol-induced intestinal permeability and liver injury [28], whereas the role of *L. bulgaricus* is poorly understood. Therefore, the present study tried to compare the prophylactic effects of oral administration of *L. bulgaricus* and *L. rhamnosus* against AFB_1_ toxicity and explore the underlying mechanisms.

In general, histopathological changes and hepatic organ index are used as indicators to evaluate the toxicity of AFB_1_ in mice [29]. Besides, AST and ALT are important markers for hepatocellular damage, as confirmed by Abdel-Moneim et al. [30]. In the present study, AST, ALT, and TP levels and hepatic organ index suggested that AFB_1_ caused liver damage, while *L. bulgaricus* or *L. rhamnosus* reversed it. Therefore, *L. bulgaricus* or *L. rhamnosus* could be useful to inhibit the progression of AFB_1_-induced hepatotoxicity, and the useful influence of *L. rhamnosus* was more obvious. It has been reported that serum levels of BUN and CREA can be employed as indices of renal dysfunction [31,32]. In the present study, decreased serum BUN and CREA activities were found in mice exposed to AFB_1_, and administration of *L. bulgaricus* led to improvements in AFB_1_-induced renal dysfunction. In contrast, the beneficial effect of *L. rhamnosus* was weak. The histological results confirmed the biochemical findings, indicating that AFB_1_ induced histological damage in the liver of mice. Similar histopathological changes of AFB_1_-induced hepatic damage have been reported in previous studies [33,34]. Furthermore, the histopathological observations of the present study, revealing that the hepatic tissues of mice treated with AFB_1_ combined with *L. bulgaricus* or *L. rhamnosus* were significantly protected, are consistent with those reported previously in mice.

Recent evidence has suggested that AFB_1_ induced the activating cytokine inflammation response in rats [35]. In our study, the cytokin response was determined in the blood and liver of mice and AML12 cells. The levels of IL-1β, TNF-α, IL-6, and IL-8 were decreased in the *L. bulgaricus*- or *L. rhamnosus*-treated group compared with the AFB_1_-treated group. A probiotic may decrease the production of TNF-α directly or indirectly by suppressing a variety of proinflammatory cytokines, such as IL 6 and IL-8 [36]. Our in vivo analyses showed decreased levels of IL-6 and IL-8, together with the decrease in the levels of proinflammatory cytokines, IL-1β and TNF-α, following the treatment of mice with *L. bulgaricus* or *L. rhamnosus*. TNF-α, a proinflammatory cytokine that can stimulate fibroblasts, endothelial cells, and macrophages to produce chemokines, causes tissue damage and chronic inflammation [37]. IL-1β is the proinflammatory cytokine which is the main index that reflects inflammatory response. The levels of IL-1β and TNF-α in culture supernatant were determined after in vitro administration of AFB_1_ and *L. rhamnosus* 1, 2, 4, and 12 h, and the results indicated the suppression of TNF-α activity, which is in agreement with the results in vivo. But in the *L. bulgaricus*-treated group, the level of TNF-α was only decreased in 2 and 4 h, which might be owing to the effects of slightly high concentrations or long culture time of *L. bulgaricus*. Taken together, these results confirmed that *L. bulgaricus* or *L. rhamnosus* treatment protected mice from the effects of AFB_1_ exposure by potently regulating cytokine production. However, the signaling pathway for the regulation of *L. bulgaricus* or *L. rhamnosus* on the expression of inflammatory cytokines remains unclear.

The mechanism of *Lactobacillus* has indicated that it has an important anti-inflammatory role in multiple signaling pathways, in which NF-κB is a key pathway. IKK complex can promote the activation of NF-κB, and when it is activated by inflammatory factors (IL-6 and IL-8, etc.), it will phosphorylate the κB signaling pathway and thereby activate NF-κB [38]. The factors secreted by *Lactobacillus casei Shirota (LcS)* can inhibit IκBα phosphorylation and degradation [39]. Live and heat-killed *Lactobacillus rhamnosus LGG* in intestinal epithelial cells can reduce IκB degradation and inhibit NF-κB translocation into the nucleus, resulting in decreased IL-8 expression [40]. In addition, *Lactobacillus reuteri* can inhibit TNF-induced p65 nuclear migration and reduces IκB degradation [41]. On the other hand, *Lactobacillus* strains are different from each other, so that further investigation of individual strains is needed to complete the mechanism by which *Lactobacillus* regulates NF-κB.

In the present study, the level of IκBα was decreased in the liver and AML12 cells treated with AFB1. Phosphorylation and proteasome-mediated degradation of IκB proteins have been reported to activate NF-κB and facilitate its transportation to the nucleus [42]. Accordingly, in the present study, AFB_1_ significantly increased the protein and mRNA expression and transcriptional activity of NF-κB. In addition, administration of *L. bulgaricus* or *L. rhamnosus* reversed the activation effects of AFB_1_ on the expression and transcriptional activity of NF-κB, indicating that *L. bulgaricus* or *L. rhamnosus* can alleviate the proinflammatory effects of AFB_1_ at the signal cascade level. However, *L. bulgaricus* treatment of 12 h had no significant effect on the restraint of this pathway. One possible explanation for this effect is that the culture time dose is too long for the adaptation of AML12 cells’ response to the stimuli. The transcription factor NF-κB plays a significant role in the activation of downstream genes responsible for secreting proinflammatory cytokines, thus mediating the levels of many proinflammatory cytokines [43,44]. The findings of the present study showed that AFB_1_ exposure resulted in inflammatory response, which was accompanied by the downregulation of NF-κB and TNF-α. Thus, *L. bulgaricus* or *L. rhamnosus* might inhibit AFB_1_-induced NF-κB activity, reducing hepatic IL-1β and TNF-α production in mice, which may be related to the suppression of inflammatory responses.

Overactivation of the NF-κB inflammatory pathway could significantly increase the expression of inflammatory cytokines [25]. In a previous study, activation of NF-κB pathway has been noted to significantly increase the TNF-α, IL-6, and IL-1β mRNA expressions in bovine hepatocytes [45]. In the present study, AFB_1_ significantly increased the mRNA expressions and concentrations of TNF-α and IL-1β, which further mediated the inflammation response of the liver in mice and AML12 cells, thereby aggravating liver or AML12 cell injury. Moreover, administration of *L. bulgaricus* or *L. rhamnosus* attenuated the effects of AFB_1_ on the mRNA expressions and concentrations of TNF-α and IL-1β, indicating that AFB_1_ induced inflammatory response via activation of the NF-κB pathway. Overall, these results indicated that AFB_1_ overactivated the NF-κB inflammatory pathway and increased the expression of inflammatory cytokines, and that *L. bulgaricus* and *L. rhamnosus* exerted a significant suppressive effect on inflammation induced by AFB_1_. Interestingly, unlike the effects of *L. bulgaricus* and *L. rhamnosus* on the NF-κB pathway, the mRNA expressions and concentrations of TNF-α and IL-1β were significantly increased following *L. bulgaricus* and *L. rhamnosus* treatment in 2 and 12 h, which might be owing to the effects of high concentrations of *L. bulgaricus* and *L. rhamnosus* and prolonged treatment duration on other inflammatory signaling pathways involved in the regulation of TNF-α and IL-1β expressions. Thus, these results emphasized the need for further investigation of the mechanism of applicable dose of *L. bulgaricus* and *L. rhamnosus* on the expression of inflammatory cytokines.

Some relevant studies demonstrated that some lactic acid bacteria can remove AFB_1_ or have protective effects against AFB_1_ [46]. Our study demonstrated that *L. bulgaricus* and *L. rhamnosus* had a protective effect against AFB_1_-induced hepatic functional damage and immune response by potently regulating cytokine production. It has been reported that *L. rhamnosus* exerts protective effects on liver and intestinal function [47]. The effects of *L. rhamnosus* can be direct, or indirect through the modulation of the endogenous flora or the immune system [48]. A previous study showed the potential of *L. rhamnosus* L60 and *L. fermentum* L23 in control of *Aspergillus* section Flavi growth and AFB_1_ production in vitro [49]. *Lactobacillus bulgaricus*, a frequent intestinal probiotic, was short of relevant studies on the anti-inflammatory action. However, our research shows that *L. bulgaricus* could inhibit the secretion of inflammatory factors by regulating the NF-κB signaling pathway, and subsequently prevent inflammatory injury caused by AFB_1_. Thus, the restraint of the NF-κB inflammatory pathway may be one of the molecular protection mechanisms of *L. bulgaricus* in mice liver caused by AFB_1_.

## 4. Conclusions

In conclusion, both in mice and in vitro, our study demonstrated that oral administration of *L. bulgaricus* or *L. rhamnosus* attenuated liver injury induced by AFB_1_ in connection with the NF-κB signaling pathway. Therefore, *L. bulgaricus* and *L. rhamnosus* administration may be a novel therapeutic approach for treating AFB1 poisoning.

## 5. Materials and Methods

### 5.1. Ethics

All procedures in the present study were performed in full compliance with the recommendation of Heilongjiang Bayi Agricultural University (Daqing, China) Institutional Animal Care and Use Committee (certification No. 2016-0019, 06 August 2017).

### 5.2. Animal Experiments

Five-week-old male Kunming mice (KM mice) were obtained from the Experimental Animal Center of Harbin Medical University (Daqing, China). After a three-day adaptation period to diet and surroundings, a total of 60 mice with similar body weight (BW) were randomly assigned to 5 groups with 12 mice in each group. Group I (control group, C) received deionized water that was used as vehicle to dissolve *Lactobacillus bulgaricus* and *Lactobacillus rhamnosus*; Group II (dimethylsulfoxide group, D) received dimethylsulfoxide that was used as vehicle to dissolve AFB_1_; Group III (AFB_1_ group, A) was treated with 300 μg/kg AFB_1_; Group IV (AFB_1_ and *L. bulgaricus* group, A+L1) was treated with 300 μg/kg AFB_1_ and 10 mg of *L. bulgaricus*; and Group V (AFB_1_ and *L. rhamnosus* group, A + L2) was treated with 300 μg/kg AFB_1_ and 10 mg of *L. rhamnosus*. There was a 12 h interval between AFB_1_ treatment and *Lactobacillus* treatment. The feeding dosage was 0.5 mL/mouse twice a day for eight weeks. Other conventional food was administered normally.

### 5.3. Sample Collection

At the end of the experiment, all the mice were fasted for 12 h. Then, blood samples were collected from the eyeballs in 1.5 mL plastic centrifuge tubes and centrifuged at 1000× *g* and 4 °C for 10 min. The serum was separated and stored at −20 °C until analysis. The mice were then sacrificed, and the liver and spleen were immediately removed. A portion of the liver and spleen were washed with ice-cold sterilized saline (0.85%), snap-frozen in liquid nitrogen, and stored at −80 °C.

### 5.4. Histopathological Examination

The liver and spleen tissues were dissected, rinsed with saline, and then fixed in 10% neutral buffered formalin. The formalin-fixed samples were routinely processed, embedded in paraffin, sectioned at 5 μm, and stained with hematoxylin and eosin. The stained sections were observed using an optical microscope, and photographs were taken.

### 5.5. Cell Culture

Mice liver cell line AML12 was purchased from the Cell Bank of Shanghai Institute of Biological Sciences at the Chinese Academy of Sciences (Shanghai, China) and cultured in Dulbecco’s modified Eagle’s medium/F12 (DMEM/F12; Invitrogen, Grand Island, NY, USA) supplemented with 10% (*v*/*v*) fetal bovine serum (Gibco, Grand Island, NY, USA), 100 U/mL penicillin, 100 U/mL streptomycin, 1% ITS Liquid Media Supplement (Gibco, Grand Island, NY, USA), and 40 ng/mL dexamethasone (Sigma, St. Louis, MO, USA) at 37 °C under an atmosphere of 5% CO_2_ in humidified air.

For dose selection experiments, the AML12 cells were treated with 0, 1, 1.2, 1.4, 1.6, 1.8, and 2 mg/mL *L. bulgaricus* and 0, 0.5, 1, 2, 3, 4, and 5 mg/mL *L. rhamnosus* for 24 h. Then, the cells were subjected to colorimetric 3-(4,5-dimethylthiazol-2-yl)-2,5-diphenyltetrazolium bromide (MTT) assay. For the in vitro experiments, the cells were treated with AFB_1_, *L. bulgaricus*, *L. rhamnosus*, and BAY 11-7082. BAY 11-7082 is an NF-κB inhibitor that inhibits the activation and translocation of NF-κB from cytoplasm into nucleus [19]. The AML12 cells were treated with 0 μg/mL AFB_1_ (Group I, C), 0.3 μL/mL DMSO (GroupII, D), 0.3 μg/mL AFB_1_ (Group III, A), 0.3 μg/mL AFB_1_ + 10 μM BAY 11-7082 (Group IV, A + B), 0.3 μg/mL AFB_1_ + 1.2 mg/mL *L. bulgaricus* (Group V, A + L1), or 0.3 μg/mL AFB_1_ + 0.5 mg/mL *L. rhamnosus* (Group VI, A + L2) for 1, 2, 4, and 12 h. Subsequently, the cell-free supernatants in each group were collected and centrifuged at 3000× rpm for 20 min.

### 5.6. Biochemical Analysis

Serum hepatic and renal biochemical indices, such as ALT, AST, ALP, TP, BUN, and CREA, were measured by personnel at the University Hospital of Jilin according to routine clinical chemistry methods (Siemens, Erlangen, Germany).

### 5.7. Determination of Parameters Associated with Inflammatory Cytokines in Liver Tissues

The serum and cell-free supernatants were separated and used to determine the concentrations of inflammatory cytokines, IL-1β, TNF-α, IL-6, IFN-γ, IL-2, and IL-8, with ELISA kit (Shanghai Lengton Bioscience Co., Road Shanghai, China), following manufacturer’s instructions.

### 5.8. Real-Time RT-PCR Analysis

Total RNA was extracted from the frozen livers with TRIzol reagent (Invitrogen Corporation, Carlsbad, CA, USA) and quantified using spectrophotometer, and its purity was assessed at 260/280 nm. Subsequently, single-stranded cDNAs were synthesized and subjected to qRT-PCR to detect the mRNA expressions of target genes. Each reaction involved preincubation at 95 °C for 3 min, followed by 39 cycles of 95 °C for 10 s, 58 °C (TA according to primer) for 30 s, and extension at 72 °C for 10 s. The relative changes in the target gene expression were determined using 2^−ΔΔCt^ method and normalized to the housekeeping gene (β-actin). The primer sequences used in this study are shown in Table 2.

### 5.9. Western Blot Analysis

The frozen livers were lysed using RIPA buffer (Beyotime Shanghai, China) containing protease inhibitors, incubated on ice for 30 min, and centrifugation at 12,000× rpm for 5 min. The protein levels in the supernatant were determined using the BCA protein assay kit (Beyotime Shanghai, China). Subsequently, equal amounts of proteins (20 μg) were separated on 10% SDS-PAGE and transferred to PVDF membranes (Millipore corp., Billerica, MA, USA). The membranes were blocked with 5% skim milk for 1 h and incubated overnight with primary antibody against p65 (#6956; 1:1000; Cell Signaling, Danvers, MA, USA), IκBα (#4814; 1:1000; Cell Signaling, Danvers, MA, USA), and β-actin (1:1000; Santa Cruz Biotechnology, Santa Cruz, CA, USA) at 4 °C. Then, the membranes were incubated with HRP-conjugated secondary antibodies (3:5000; Beyotime Shanghai, China) for 30 min at room temperature. The protein expression signals were visualized by ECL (Beyotime Shanghai, China), and band intensity was quantified using the Image J software (NIH, Bethesda, MD, USA).

### 5.10. Statistical Analysis

The data were analyzed with SPSS 17.0, and are presented as mean ± standard deviation. Statistical significance was evaluated by one-way analysis of variance (ANOVA) with Duncan test for post hoc analysis. Differences with *p* < 0.05 and *p* < 0.01 were considered statistically significant and highly significant, respectively.

## Figures and Tables

**Figure 1 toxins-11-00017-f001:**
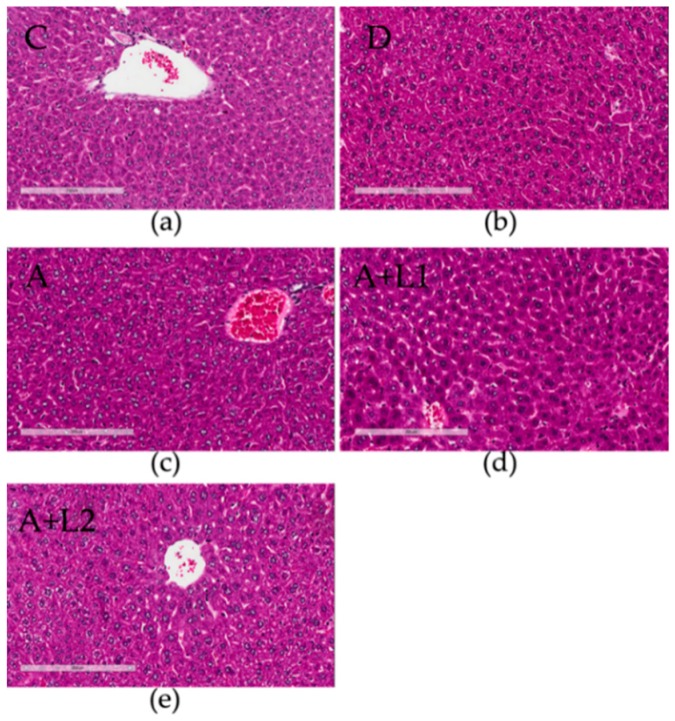
Histopathological observation of livers after different treatments. Typical images were chosen from each experimental group (original magnification: 200×): (**a**) Control group. (**b**) Vehicle group (DMSO). (**c**) Mice treated with AFB1. (**d**) Mice treated with AFB1 and *L. bulgaricus*. (**e**) Mice treated with AFB1 and *L. rhamnosus*.

**Figure 2 toxins-11-00017-f002:**
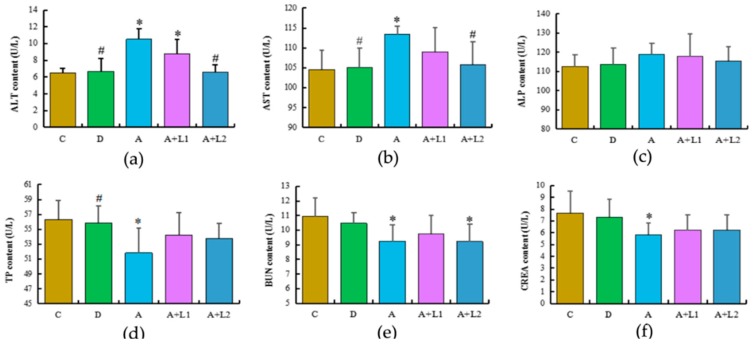
Effects of *L. bulgaricus* or *L. rhamnosus* on serum (**a**) ALT, (**b**) AST, (**c**) ALP, (**d**) TP, (**e**) BUN, and (**f**) CREA levels in mice exposed to AFB1. Each value represents the mean ± SD. * Denotes significant differences (*p* < 0.05) between the control group and other groups. ^#^ Indicates significant differences (*p* < 0.05) between the AFB1-treated group and AFB1 + *Lactobacillus* treatment groups.

**Figure 3 toxins-11-00017-f003:**
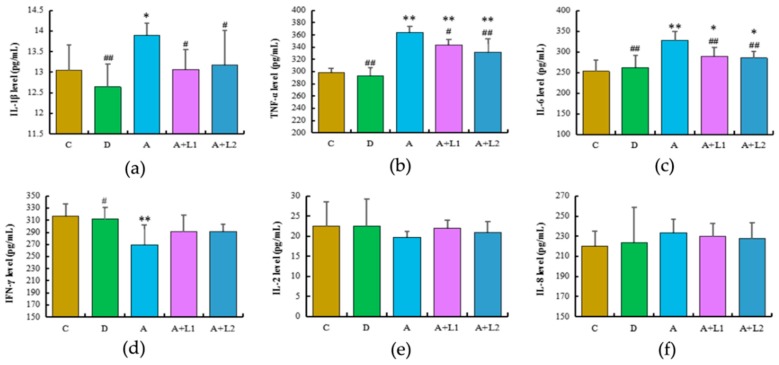
Effects of *L. bulgaricus* or *L. rhamnosus* on serum (**a**) IL-1β, (**b**) TNF-α, (**c**) IL-6, (**d**) INF-γ, (**e**) IL-2, and (**f**) IL-8 levels in mice exposed to AFB_1_. Each value represents mean ± SD. * and ** denote significant differences (*p* < 0.05 and *p* < 0.01, respectively) between the control group and other groups. ^#^ and ^##^ indicate significant differences (*p* < 0.05 and *p* < 0.01, respectively) between the AFB_1_-treated group and AFB_1_ + *Lactobacillus* treatment groups.

**Figure 4 toxins-11-00017-f004:**
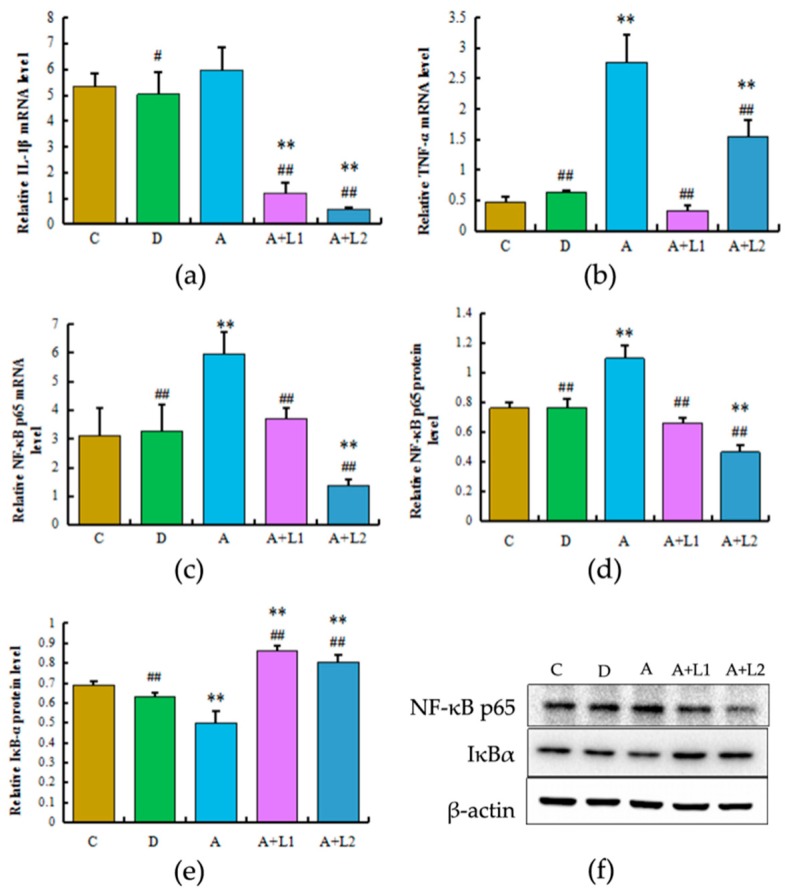
Effects of *L. bulgaricus* or *L. rhamnosus* on (**a**) IL-1β, (**b**) TNF-α, and (**c**) NF-κB p65 mRNA expression levels as well as (**d**,**f**) NF-κB p65 and (**e**,**f**) IκBα protein expression induced by AFB_1_ in the liver. Each value represents mean ± SD. * and ** denote significant differences (*p* < 0.05 and *p* < 0.01, respectively) between the control group and other groups. # and ## indicate significant differences (*p* < 0.05 and *p* < 0.01, respectively) between the AFB_1_-treated group and AFB_1_ + *Lactobacillus* treatment groups.

**Figure 5 toxins-11-00017-f005:**
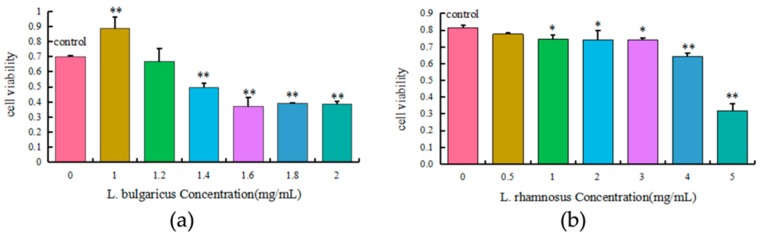
Effects of various concentrations of (**a**) *L. bulgaricus* or (**b**) *L. rhamnosus* on the viability of AML12 cells. Each value represents the mean ± S.D. * and ** denote significant differences (*p* < 0.05 and *p* < 0.01, respectively) between the control and other groups.

**Figure 6 toxins-11-00017-f006:**
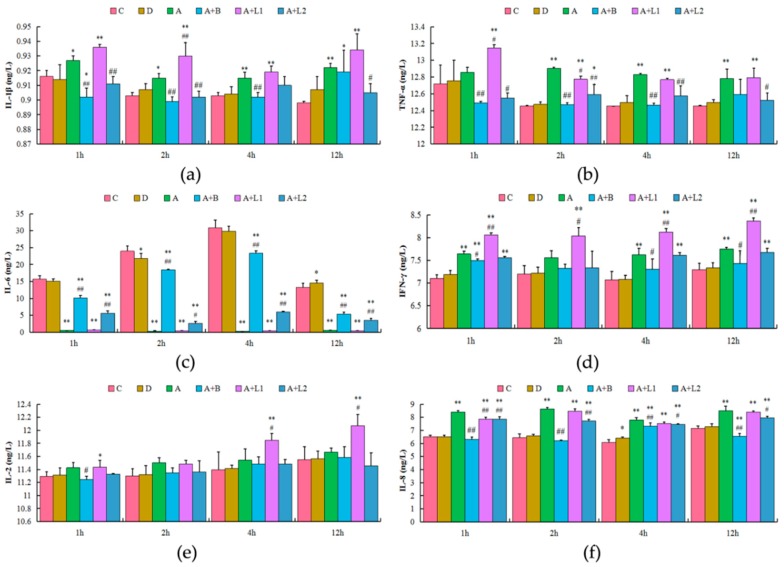
Effects of *L. bulgaricus* or *L. rhamnosus* on the (**a**) IL-1β, (**b**) TNF-α, (**c**) IL-6, (**d**) INF-γ, (**e**) IL-2, and (**f**) IL-8 levels in AML12 cells exposed to AFB_1_. Each value represents mean ± S.D. * and ** indicate significant differences (*p* < 0.05 and *p* < 0.01, respectively) between the control (without DMSO) and other groups. # and ## denote significant differences (*p* < 0.05 and *p* < 0.01, respectively) between the AFB_1_-treated group and AFB_1_ + *Lactobacillus* treatment groups.

**Figure 7 toxins-11-00017-f007:**
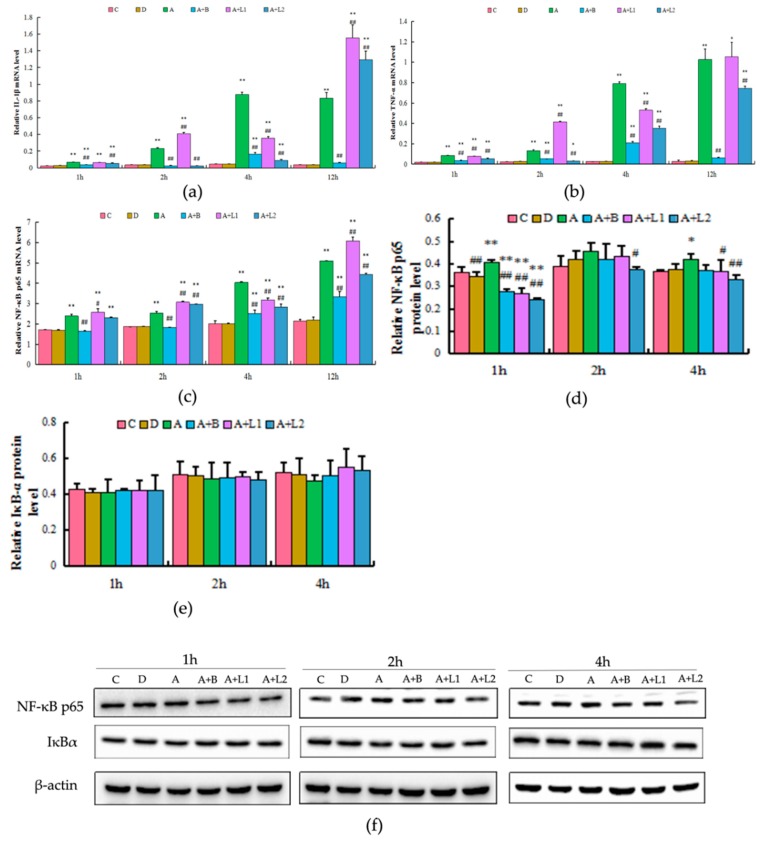
Effects of *L. bulgaricus* or *L. rhamnosus* on the (**a**) IL-1β, (**b**) TNF-α, and (**c**) NF-κB p65 mRNA expression and (**d**,**f**) NF-κB p65 and (**e**,**f**) IκBα protein expression induced by AFB_1_ in AML12 cells. Each value represents mean ± S.D. * and ** denote significant differences (*p* < 0.05 and *p* < 0.01, respectively) between the control group and other groups. ^#^ and ^##^ indicate significant differences (*p* < 0.05 and *p* < 0.01, respectively) between the AFB_1_-treated group and AFB_1_ + *Lactobacillus* treatment groups.

**Table 1 toxins-11-00017-t001:** Body weights of mice measured every week.

Groups	Body Weights (g)								Liver Index (%)
	Day 7	Day 14	Day 21	Day 28	Day 35	Day 42	Day 49	Day 56	
C	36.04 ± 0.54	37.07 ± 3.17	37.69 ± 3.72	41.53 ± 2.10	42.83 ± 1.88	43.74 ± 2.82	45.93 ± 2.32	46.19 ± 3.31	4.02 ± 0.243
D	36.57 ± 1.03	38.53 ± 1.99	39.73 ± 1.17	42.20 ± 1.61	43.63 ± 2.84	43.83 ± 2.34	46.27 ± 2.48	45.47 ± 1.50	3.96 ± 0.189 ^#^
A	35.47 ± 1.14	41.02 ± 1.62 *	39.82 ± 1.48	39.63 ± 2.79	41.05 ± 4.28	41.20 ± 4.84	44.17 ± 1.96	45.47 ± 1.86	4.27 ± 0.171 *
A + L1	36.40 ± 2.25	36.16 ± 2.97 ^#^	36.58 ± 3.21 ^#^	40.33 ± 4.96	42.79 ± 2.70	43.66 ± 3.91	44.53 ± 3.12	46.47 ± 2.29	4.12 ± 0.191
A + L2	36.70 ± 2.80	35.05 ± 2.02 ^#^	39.35 ± 1.82	39.30 ± 2.68	38.38 ± 3.52 *	42.00 ± 1.90	43.43 ± 2.80	45.35 ± 2.80	4.09 ± 0.168

^1^ Each value represents mean ± SD. * Denotes significant differences (*p* < 0.05) between the control group and other groups. ^#^ Indicates significant differences (*p* < 0.05) between the AFB1-treated group and AFB1 + *Lactobacillus* treatment groups.

**Table 2 toxins-11-00017-t002:** Primer sequences used in this study.

Gene	Primer Sequence (5′→3′)	Product Size (bp)
β-actin	F: 5′-GAGACCTTCAACACCCCAGC-3′	263
	R: 5′-ATGTCACGCACGATTTCCC-3′	
p65	F: 5′-GCTCCTGTTCGAGTCTCCATG-3′	91
	R: 5′-CATCTGTGTCTGGCAAGTACTGG-3′	
IL-1β	F: 5′-AGCTTCAAATCTCGCAGCAG-3′	72
	R: 5′-TCTCCACAGCCACAATGAGT-3′	
TNF-α	F: 5′-CTCATGCACCACCATCAAGG-3′	96
	R: 5′-ACCTGACCACTCTCCCTTTG-3′	

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
