# Peer review of "Lactobacillus bulgaricus or Lactobacillus rhamnosus Suppresses NF-κB Signaling Pathway and Protects against AFB1-Induced Hepatitis: A Novel Potential Preventive Strategy for Aflatoxicosis?"

_toxins, 2019, doi:10.3390/toxins11010017_

Round 1
Reviewer 1 Report
The manuscript is well written and clear, the experiment adequately conducted and results properly commented. Authors explained the process precisely and related data were well organized and the topic is highly relevant. It has been a pleasure for me to read this manuscript and I will strongly suggest to publish the present study in ‘’Toxins’’

Author Response
6 December, 2018
Dear Reviewer,
Re: Revised Manuscript toxins-393703
On behalf of all co-authors, I would like to thank you and the reviewer for favorable comments and constructive suggestions on our manuscript (MS) toxins-393703, which are very valuable for us to improve the MS. The reviewer liked our MS, considered that it is of interest to the readership of Toxins. The reviewer are very kind and generous to have provided detailed constructive comments and suggestions for us to improve the quality of the MS. We have revised the MS by strictly according to the reviewer’ comments and suggestions. The changed text and sentences are shown in red in the revised MS. In the following, we detail our point-by-point responses to these specific comments and suggestions.
Response to Reviewer 1 Comments
Point 1: The manuscript is well written and clear, the experiment adequately conducted and results properly commented. Authors explained the process precisely and related data were well organized and the topic is highly relevant. It has been a pleasure for me to read this manuscript and I will strongly suggest to publish the present study in “Toxins”
Response 1: We thank the Reviewer 1 very much for favorable comments on our MS.
We have carefully examined the manuscript and corrected other mistakes and typo-errors.
We sincerely hope that the MS has been revised satisfactorily.We are looking forward to seeing the acceptance of the revised manuscript for publication in Toxins as soon as possible.
Kindest regards

Reviewer 2 Report
The article presents a novel and thorough evaluation of a practical method for aflatoxin toxicity mitigation. The article is concise and well written. Only a couple of minor suggestions.
Line 8-9 the authors state in the "To overcome aflatoxin toxicity, probiotic-mediated detoxification, in which probiotics bind 8 to aflatoxin and prevent its absorption in the small intestine, has been proposed." This statement seems out of place. Although binding is one of the proposed pathways of aflatoxin deactivation in this study authors are evaluating a different pathway. I don't think this sentence adds value and it may lead a reader to think that the authors are going to evaluate this mode of action, which they don't. I recommend deleting this sentence.
Line 27 the word should be "protective"
Line 28 a space is missing between by^AFB1
Line 39 - contamination is a better term change pollution to contamination.
Line 59-60 - authors state "which are the most susceptible to AFB1 poisoning" in reference to chickens. Of the researched species both Turkey and ducks are more susceptible than chickens. I would recommend the authors state something in the order of "chickens which are the most susceptible of the agriculturally important species" or just simply say "one of the most susceptible species"
Author Response
6 December, 2018
Dear Reviewer,
Re: Revised Manuscript toxins-393703
On behalf of all co-authors, I would like to thank you and the reviewer for favorable comments and constructive suggestions on our manuscript (MS) toxins-393703, which are very valuable for us to improve the MS. The reviewer liked our MS, considered that it is of interest to the readership of Toxins. The reviewer are very kind and generous to have provided detailed constructive comments and suggestions for us to improve the quality of the MS. We have revised the MS by strictly according to the reviewer’ comments and suggestions. The changed text and sentences are shown in red in the revised MS. In the following, we detail our point-by-point responses to these specific comments and suggestions.
Response to Reviewer 2 Comments
The article presents a novel and thorough evaluation of a practical method for aflatoxin toxicity mitigation. The article is concise and well written. Only a couple of minor suggestions.
Point 1: Line 8-9 the authors state in the "To overcome aflatoxin toxicity, probiotic-mediated detoxification, in which probiotics bind 8 to aflatoxin and prevent its absorption in the small intestine, has been proposed." This statement seems out of place. Although binding is one of the proposed pathways of aflatoxin deactivation in this study authors are evaluating a different pathway. I don't think this sentence adds value and it may lead a reader to think that the authors are going to evaluate this mode of action, which they don't. I recommend deleting this sentence.
Response 1: We thank the Reviewer 2 very much for constructive comments and suggestions on our MS, and we have revised this sentence.
To overcome aflatoxin toxicity, probiotic-mediated detoxification has been proposed.
Point 2: Line 27 the word should be "protective"
Response 2: We thank the Reviewer 2 very much for constructive comments and suggestions on our MS, and we have revised this sentence.
However, its protective effect and anti-inflammatory mechanism induced by AFB1 remains largely unknown.
Point 3: Line 28 a space is missing between by^AFB1
Response 3: We thank the Reviewer 2 very much for constructive comments and suggestions on our MS, and we have revised this sentence.
See point 2
Point 4: Line 39 - contamination is a better term change pollution to contamination.
Response 4: We thank the Reviewer 2 very much for constructive comments and suggestions on our MS, and we have revised this sentence.
In China, AF contamination generally refers to AFB1 contamination.
Point 5: Line 59-60 - authors state "which are the most susceptible to AFB1 poisoning" in reference to chickens. Of the researched species both Turkey and ducks are more susceptible than chickens. I would recommend the authors state something in the order of "chickens which are the most susceptible of the agriculturally important species" or just simply say "one of the most susceptible species"
Response 5: We thank the Reviewer 2 very much for constructive comments and suggestions on our MS.
A fact we want to stated in this sentence is that lactic acid bacteria could alleviate the degree of liver inflammation and hepatic damage, and exert a positive effect on the prevention of AFB1 poisoning in chickens, which has been proven in our previous studies. And we have revised this sentence.
Our previous studies have shown that lactic acid bacteria could alleviate the degree of liver inflammation and hepatic damage, and exert a positive effect on the prevention of AFB1 poisoning in chickens, which are one of the most susceptible to AFB1 poisoning.
We have carefully examined the manuscript and corrected other mistakes and typo-errors.
We sincerely hope that the MS has been revised satisfactorily.We are looking forward to seeing the acceptance of the revised manuscript for publication in Toxins as soon as possible.
Kindest regards
